# Psychometric Properties of Interpersonal Regulation Questionnaire for Chinese College Students: Gender Differences and Implications for Well-Being

**DOI:** 10.3390/bs13060507

**Published:** 2023-06-16

**Authors:** Yanhua Zhao, Niu Wang, Jiahui Niu, Xingchen Li, Lei Zhang

**Affiliations:** 1School of Psychology, Henan University, Jinming Campus, Kaifeng 475001, China; yz@vip.henu.edu.cn (Y.Z.); wn@henu.edu.cn (N.W.); njh@henu.edu.cn (J.N.); 2School of Computer and Information Engineering, Henan University, Jinming Campus, Kaifeng 475001, China; lixingchen@henu.edu.cn

**Keywords:** Interpersonal Regulation Questionnaire, gender differences, social and emotional well-being, Chinese college students

## Abstract

Intrapersonal emotion dysregulation has been found to be a transdiagnostic predictor in the development of almost all affective disorders. Interpersonal resources are also involved in achieving people’s emotion regulation goals. The Interpersonal Regulation Questionnaire (IRQ) has been developed to assess the tendency and efficacy of people using external resources to help manage their emotions. Under the restrictions of the COVID-19 pandemic, the role of interpersonal emotion regulation in individuals’ adjustment and well-being remains unclear. This study aimed to investigate the optimal factor structure of the IRQ in Chinese culture using an exploratory structural equation modeling approach and to examine the associations between interpersonal emotion regulation, tested by the IRQ, and young people’s intrapersonal emotion dysregulation and social and emotional well-being. The sample consisted of 556 college students aged from 17 to 31 from Mainland China. Factor analyses suggested that the four-factor structure was the optimal model for the current data. Females reported a higher tendency to use external resources to regulate their negative emotions and higher efficacy in regulating negative emotions. The Chinese version of the IRQ (C-IRQ) presented adequate psychometric properties and would be a useful tool for measuring interpersonal emotion regulation behaviors.

## 1. Introduction

It is not unusual to seek comfort from family or friends when one’s mood is unstable. The simple act of holding important others’ hands can down-regulate people’s physiological responses to external stressors [1,2]. People can not only regulate their own emotions independently, but they can also achieve emotion-regulatory goals through the process of social interaction [3,4,5]. Zaki and Williams provided a process model of interpersonal emotion regulation (IER), which divided IER into intrinsic IER (regulating personal emotions by connecting oneself to other people) and extrinsic IER (regulating others’ emotions) [6]. Although most emotions are generated from the social interaction process and modulated interpersonally [3,4], the current literature places more emphasis on the intrapersonal aspects of emotion regulation [7,8].

To regulate one’s own emotions, the IER highlights the process of pursuing emotional goals by seeking help from others [6,9]. Individuals often benefit from social support in situations demanding emotion regulation [10,11,12,13]. Help-seeking behaviors have presented a positive impact on alleviating distress [14,15]. The IER process, which involves the use of social resources to manage emotions, has demonstrated significant associations with people’s social relationships, well-being indicators, and psychopathology [9,16,17,18]. Although theorists have pointed out the importance of IER, few empirical studies have investigated the impact of IER on people’s important outcomes.

### 1.1. Tools Measuring Intrinsic IER

One important reason why the field has not been extensively studied may be the lack of high-quality tools to assess the components of interpersonal emotional regulation [17]. Several researchers have attempted to measure the process of using interpersonal interaction to improve intrapersonal emotional states. Hofmann and his colleagues developed a 20-item scale called the IER Questionnaire (IERQ) to measure how people use interpersonal resources to regulate their emotions (e.g., perspective-taking) [17]. Individuals who experience greater negative affect tend to report a higher frequency of using IER strategies to regulate their emotions [17]. 

Recently, based on the IER model of Zaki and Williams [6], the Interpersonal Regulation Questionnaire (IRQ) has been developed to assess the tendency and efficacy of people using IER to manage their emotions [9]. The IRQ mainly includes two dimensions, IER *tendency* (the tendency of people to use IER during emotional events) and *efficacy* (the perceived efficacy of using IER to improve personal emotions). Combined with the affective goals of decreasing negative and increasing positive emotional experiences, the IRQ could assess individual differences in four aspects of IER: Negative Tendency (NT), Negative Efficacy (NE), Positive Tendency (PT), and Positive Efficacy (PE). *NT* refers to the tendency to seek other people’s help for intrapersonal emotion regulation in response to negative emotional events. *NE* refers to the efficacy people perceived after obtaining external help in improving their negative emotions. *PT* refers to the tendency to share with other people in response to positive emotional events. *PE* refers to the efficacy people perceive in sharing their positive emotions with other people. From the initial validation of the IRQ [9], the four subscales have satisfied reliability and validity in connecting with individuals’ social (i.e., social anxiety, prosociality, and loneliness) and emotional well-being indicators (i.e., positive and negative affect and perceived stress). Further studies investigating the structure of the IER would provide greater empirical support for the validity of the IER model. 

### 1.2. Factor Structure of IRQ

Regarding the factors’ structure, the developers of the IRQ have attempted to classify the four subdimensions of the intrinsic IER alongside two higher-ranking conceptions (*Tendency* and *Efficacy*), with both positive and negative emotional situations. The original validation study supports the proposed four-factor structure of IRQ (NT, NE, PT, and PE) [9]. Although the designers initially regarded *Tendency* and *Efficacy* as the key dimensions of IER, these higher-order factors are not examined in this study. Thus, we want to examine whether *Tendency* and *Efficacy* could be represented by the four factors. Moreover, we also want to explore the two emotional situations of the IRQ, namely, *Positive* and *Negative*. Watson and Tellegen found that positive emotion and negative emotion are the main dimensions of emotional experience through the analysis of many studies on self-rated emotion [19]. These dimensions are widely used in self-reported emotion research. Therefore, the question of whether the four factors of the IRQ can represent the higher-order structure of positive and negative emotional situations is due for exploration.

A recent study examined the original four-factor structure, two-factor structures, and the possible hierarchical factor models for the IRQ using a Chinese adolescent sample [20]. The findings suggest that the four-factor model best represented the adolescent data. However, the results also showed that the between-factor correlations were relatively high (r = 0.49–0.77, mean*_r_* = 0.63) compared with the original study (r = 0.43–0.65, mean*_r_* = 0.52), implying a possible hierarchical factor underlying the whole structure. The higher-order model and the bifactor model representing a general underlying factor were not selected as the best model for this adolescence data because of several concerns regarding model fitting, leaving the optimal structure of the IRQ undetermined.

### 1.3. IER and Social and Emotional Well-Being in Chinese Youths

Interpersonal relationships, meaning the interdependence and interaction between individuals, profoundly affect people’s lives in almost all areas of Chinese culture [21]. Emotion is an integral part of defining interpersonal relationships in the Chinese language [22]. Using interpersonal interaction to help regulate one’s personal emotions is very common in Chinese culture. East Asian people tend to report higher habitual use of IER strategies (social modeling and perspective-taking) in daily life than Western people [23]. However, it remains unclear how the habitual use of IER contributes to individuals’ important outcomes [24,25]. 

Several studies have attempted to associate IER with individuals’ social adjustment outcomes. For example, individuals’ attempts to adjust their emotions in a relationship situation have an important impact on the quality of social relationships [26]. Using IER for perspective-taking may increase the quality of social relationships [27]. IER efficacy and tendency are positive predictors of prosociality, social connectedness, and low levels of loneliness and social anxiety [9]. Using a Chinese adolescent sample, Ding et al. and Chen et al. present similar findings which associate IRQ scales with a higher level of prosociality and social initiative behaviors and lower-level social withdrawal behaviors [20,28]. Nevertheless, the impact of interpersonal behaviors on social outcomes is not always positive. The use of IRQ for soothing and social modeling could increase interpersonal relationship problems [29] and social anxiety [30].

Moreover, the results regarding the relationship between IER and emotional well-being are inconsistent. Some studies showed that using some IER strategies (e.g., enhancing positive affect) could decrease internalizing symptoms [31], and the tendency or efficacy of using IER may increase positive affect [9]. Some studies showed that the use of IER to regulate emotions was positively related to high internalizing symptoms and lower levels of general well-being [17,27,32], especially using soothing [27,32] or other maladaptive IER strategies for IER [24]. However, under certain circumstances (people using less maladaptive emotional regulation strategies), some IER strategies (soothing) could also reduce depression [33], and in other cases (when people use low-level inner resources in regulating their emotions), IER strategies (perspective-taking) may increase the level of depression. In studies using Chinese youth samples, the use of IER has been associated positively with positive emotions and negatively with negative emotions or distress [20,28,34]. In summary, these results suggest that the effects of IER on social and emotional well-being are mixed, and the reasons for this are worthy of further exploration.

### 1.4. The Present Study

Existing research suggests that IER styles are potential predictors of social and emotional well-being [9,17,24,27]. Based on the theoretical and empirical importance, in this study, we aimed to examine the construct validity of the Chinese version of the IRQ for a Chinese youth sample. First, we investigated whether the factor structure of the IRQ in the Chinese sample is consistent with that in American samples. Based on previous studies, we compared five competitive models of the IRQ, including a four-factor model, a model with a higher-order factor, models with two higher-order factors, and a bifactor model. Second, studies comparing gender differences in IER have shown that women use some IER behaviors more frequently than men. Thus, the latent gender differences were compared based on the measurement equivalence test. Then, we established the convergent validity of the scale by connecting IRQ scales with extraversion personality, which represents some social sharing and interaction behaviors as IER [9]. The discriminant validity was established by connecting the IRQ scales with emotional dysregulation levels, which assess individuals’ intrapersonal emotion regulation resources. Existed studies have shown that IER is positively related to individuals’ difficulties in self-regulating their emotions [17,27,29,34]. We also examined the relationship patterns between the IRQ scales and meaningful social and emotional indicators, comparing these with those obtained from previous findings [9,27]. 

## 2. Methods

### 2.1. Participants and Procedure 

Before data collection, the power and sample size were estimated based on RMSEA proposed by MacCallum et al. (1996) [35]. To estimate the minimum required sample size for this study, the degrees of freedom were calculated for all proposed models (with the number of degrees of freedom ranging from 104 to 50); the ideal sample size would range from 135 to 207 to achieve 90% power to reject the hypothesis of not-close fit (RMSEA ≥ 0.05) at a 5% significance level [36]. Considering the gender equivalence of these models, slightly larger sample sizes are required. A total of 556 participants (346 females) were recruited among university students in Henan, Mainland China. Participants ranged in age from 17 to 31 (age: M = 19.59 years, SD = 2.133, 66% 17–19 years old, 32.6% 20–24 years old, 3.8% 25–31 years old). Among them, 96.4% were Han People, and 3.6% were the minority. A total of 301 students participated in a long paper questionnaire survey (demographic information and all measures mentioned in the *Measures* section) lasting approximately 20 min. The remaining 255 students completed a short questionnaire survey (demographic information and IRQ), taking approximately five minutes. 

Participants were randomly selected in playgrounds, restaurants, libraries, etc., on a university campus. Students who agreed to participate received a questionnaire package. Participants were informed about the voluntary and confidential nature of the research and that they could exit the survey at any time. The study was approved by the Medical and Scientific Research Ethics Committee of the authors’ university.

### 2.2. Measures

*Interpersonal Regulation Questionnaire (IRQ)*. The IRQ is a self-report scale that evaluates individuals’ behavioral tendencies and efficacy in seeking IER, and was designed by Williams et al. [9]. It comprises 16 items, represented by 4 dimensions: NT (4 items), NE (4 items), PT (4 items), and PE (4 items). Participants rated each item on a 7-point Likert scale from 1 (*strongly disagree*) to 7 (*strongly agree*). The IRQ has good reliability and validity, and the alpha reliability coefficients of the subscales were 0.83–0.90 [9]. The Chinese version of IRQ (C-IRQ) is set through a process of translation and back-translation. The back-translated items are compared with the original scale items. In the case of inconsistencies between these items, the translated Chinese items were revised until the inconsistencies were eliminated. 

*Perceived Extraversion.* The extraversion subscale was selected from the Chinese Version of the Ten-Item Personality Inventory published by Lu et al. [37], translated from the original English version of Gosling et al. [38]. Two items, including “extraverted, enthusiastic” and “reserved, quiet,” were used to assess the extraversion dimension from the Big Five personality structure. Participants gave ratings using a scale from 1 (*strongly disagree*) to 6 (*strongly agree*). The correlation between the two items was 0.54.

*Emotion Dysregulation.* The Chinese Version of Difficulties in Emotion Regulation Scale (DERS) was validated by Zhao et al. [39] as a brief form of Gratz and Roemer’s DERS [40]. The 15-item scale, comprising 5 subscales (including clarity, goals, non-acceptance, impulse, and strategies), was used to measure students’ difficulties in regulating their emotions. These items are rated on a 5-point Likert scale ranging from 1 (*almost never*) to 5 (*almost always*). Higher total scores on the scale indicated higher levels of students’ emotion dysregulation. The Cronbach’s alpha for DERS-15 was 0.90. 

*Perspective-Taking and Empathetic Concern*. Two subscales, including Perspective-Taking (5 items) and Empathetic Concern (5 items), were selected from the Chinese Version of the Interpersonal Reactivity Index published by Zhang et al. [41] to assess individuals’ empathetic ability in our study. Answers were recorded on a 5-point scale, ranging from 1 (*inappropriate*) to 5 (*very appropriate*). The Cronbach’s alpha coefficient was 0.67 for Empathetic Concern in this study. The original Cronbach’s alpha coefficient was 0.60 for Perspective-Taking. After screening the items, the item “If I’m sure I’m right about something, I don’t waste much time listening to other people’s arguments” was deleted from Perspective-Taking because of its low item-total correlation, which improves the Cronbach’s alpha of this subscale to 0.67. 

*Social Anxious behavior*. The Fear of Social Interaction (FSI) subscale was selected from He and Zhang’s Chinese version of the Liebowitz Social Anxiety Scale [42] to assess students’ socially anxious behaviors. The FSI addresses 11 social interactional situations with 11 items (e.g., meeting strangers); participants were asked to rate their fear of these situations on a 4-point scale from 1 (*not at all*) to 4 (*very much*). The Cronbach’s alpha for the FSI was 0.85. 

*Loneliness*. The Three-Item UCLA Loneliness Scale (UCLA-3) was developed by Hughes et al. [43] to measure feelings of loneliness. The scale includes 3 items, ranging from 1 (*never*) to 4 (*often*), on a 4-point Likert scale. The higher the score, the higher the feelings of lack of company, of being forgotten, and of isolation will be. Although there is no report on the use of UCLA-3 in the Chinese sample, the longer version of UCLA has been reported to have good reliability and validity [44]. The one-factor structure of UCLA-3 was confirmed by running a confirmatory factor analysis, which resulted in an excellent model fit, *χ*^2^ =2.15, *df* = 1, *p* = 0.14; CFI = 0.998, TLI = 0.994, RMSEA = 0.062, SRMR = 0.01. The Cronbach’s alpha for the scale was 0.72.

*Positive and Negative Affect*. The Chinese Version of Positive and Negative Affect Scale was validated by Miao [45] to assess the positive and negative emotional states of individuals in the past 1–2 weeks. The scale consisted of 12 questions, including Positive Affect (6 items) and Negative Affect (6 items). Participants were asked to rate their recent emotional experiences on a 5-point Likert scale, with emotional adjectives ranging from 1 (*at no time*) to 7 (*all the time*). The Cronbach’s alpha for the subscale was 0.81 for Positive Affect and 0.74 for Negative Affect.

*Depressive Symptoms.* The depressive symptoms scale was obtained from Gong and her colleague’s Chinese Version of the Depression, Anxiety, and Stress Scale (DASS-21) to assess students’ depressive symptoms [46]. Participants rated each of 7 items on a 5-point scale from 1 (*did not apply to me at all*) to 4 (*applied to me most of the time*). The Depression scale has good reliability and validity in Chinese student samples [46]. The Cronbach’s alpha of the Depression scale was 0.86 in this study.

### 2.3. Data Analyses

We use exploratory structural equation modeling (ESEM) to test the proposed correlated four-factor, higher-order factor, the bifactor models, and models with two higher-order factors. ESEM integrates the functions and advantages of both EFA and CFA analysis methods [47]. All models were estimated with Mplus Version 8 [48]. After screening the items of the IRQ, we found that the data were multivariate and non-normally distributed. The data had missing values ranging from 0.3% to 1.7% for the IRQ items. We thus selected the MLR estimator (a maximum likelihood estimation with robust standard errors and a scaled test statistic) available in Mplus 8 to handle the non-normality and missing value issues for model analyses. We selected three commonly used fit indices to determine the fit of models: the comparative fit index (CFI), the Tucker–Lewis index (TLI), and the root mean square error of approximation (RMSEA). According to Hu and Bentler [49], CFI and TLI values of 0.95 or greater reflect an excellent model fit, and an RMSEA value of 0.06 or less reflects a good fit to the data. In model comparison and selection, ΔCFI ≤ 0.01 and ΔRMSEA ≤ 0.015 were considered non-significant changes in the model fit [50].

## 3. Results 

### 3.1. Factor Analyses

As recommended by Marsh and colleagues [47], ESEM with targeted rotation was used for confirmatory purposes. The four first-order factors model was selected as the priority factor structure of the IRQ (see Figure 1). 

This model (M0) resulted in a satisfied model fit, S-B scaled *χ*^2^ = 183.10, *df* = 62, *p* < 0.01; CFI = 0.95, RMSEA = 0.06 (0.05–0.07), SRMR = 0.03. The membership of each item assigned in ESEM, standardized item factor loadings (ranging from 0.40 to 0.84), and the latent correlation between four factors are presented in Table 1. For comparison purposes, we also estimated the four-factor CFA model and a higher-order model with four first-level factors for IRQ (see Appendix A), which resulted in a poor model fit (see Appendix A).

Five alternative models were structured (see Figure 2) and estimated (see Table 2). As a fundamental idea to construct a one-factor concept model, we estimated a general-factor model (M1) with all items loaded on one latent factor. The findings showed that the model was not acceptable, indicating the applicability of a more complex factor structure. To estimate whether the IRQ items could be explained by a hierarchical factor, we examine a higher-order factor model using the procedure of ESEM within CFA (M2). The results showed a good model fit. The factor loadings of four first-order factors ranged from 0.27 to 0.93. Because the factor loading of NT was lower than the accepted value of 0.30 for constructing a higher-order factor, this higher-order model was not selected. Meanwhile, a bi-factor ESEM model was estimated based on a previous proposal [20] that the four different dimensions of IRQ may be explained by a single underlying factor in addition to the four specific factors (M3). Although this model had a significantly better fit than the four-factor models, the relatively low composite reliability of the general factor (McDonald Omega Hierarchical = 0.68), which was lower than the cutoff value 0.80 proposed by Reise, Bofani, Haviland [51], suggested that the bi-factor model was not preferable. In this bi-factor model, the loading of item 5 on the general factor was not significant, the loading of item 3 on specific factor 1 was negative, the loading of item 4 on specific factor 1 was also small, and those non-significant, negative, and small factor loadings indicate an anomalous and inadmissible model specification [52]. Item loadings for the general and specific factors in the bifactor model are presented in Table 3. 

Next, a higher-order factor model with *Efficacy* and *Tendency* and another with *Positive* and *Negative* as higher-order dimensions were estimated to examine whether the higher-order factors (*Efficacy* and *Tendency*, *Positive* and *Negative*) could be supported, as designed by Williams et al. [9]. The results indicated that the model with *Efficacy* and *Tendency* dimensions fit the data well. However, after screening the factor loadings, we found that the factor loadings of NT and PT were lower than 0.30, which was insufficient to construct a factor, *Tendency*. Moreover, the results indicated that the higher-order factor model with *Positive* and *Negative* also fit the data well. Nevertheless, factor loadings for the factor *Negative* were not sufficient (loadings lower than 0.30; see Table 3). Thus, these models with two higher-order factors were not acceptable for the current data. Finally, the four-factor ESEM model was retained for further analyses. 

### 3.2. Measurement Invariance and Latent Gender Differences

Measurement invariance was then examined by comparing the models with and without equality constraints on model parameters (see Table 2). First, a configural invariance model (with no equality constraints) was tested, which resulted in a good model fit.

Then, a metric invariance model with all factor loadings being equally constrained across gender was tested and compared with the configural invariance model. The metric invariance model fit the data well and did not significantly differ from the configural model (ΔCFA < 0.01, ΔRMSEA < 0.015), supporting further testing of the scalar invariance model. Finally, a scalar invariance model with constrained item intercepts was specified. The scalar invariance model had an acceptable model fit and was not significantly worse compared to the metric invariance model (ΔCFA < 0.01, ΔRMSEA < 0.015), supporting the scalar invariance model. In summary, these findings indicate the suggested four-factor model fits the data equally well across men and women.

Based on the measurement equivalence test, significant latent mean differences across gender were detected (see Appendix A). The latent factor means were set at 0 in the male group and were allowed to vary in the female group. The estimated parameters of the female group represent the difference in latent means across gender groups. As shown in Appendix A, females reported significantly higher NT and NE values than males, indicating that in facing negative emotional situations, females are more likely to exert external resources to manage their emotions and achieve higher efficacy in doing this.

### 3.3. Internal Consistency Reliability

The Cronbach’s alpha coefficient of the whole IRQ was 0.89, and for the four subscales, was 0.78 for NT, 0.80 for NE, 0.83 for PT, and 0.82 for PE, respectively. The internal consistency reliability coefficients of all subscales were higher than 0.70. Items’ corrected item-total correlations within factors were above 0.51, and the deletion of any item could not improve the internal consistency reliability. Thus, all the original items were retained to capture the phenomenon of IER in Chinese culture.

### 3.4. Validity Analyses

We next examined the convergent and discriminant validity of the IRQ scale by connecting its subscales with extraversion and intrapersonal emotion dysregulation. As for extraversion, PT and PE marginally correlated with extraversion. NT, NE, and PT demonstrated small to medium positive relationships with intrapersonal emotion dysregulation.

The relationship between the mean scores of the IRQ subscales and a series of indicators of social (Empathetic Concern, Perspective-Taking, Anxious Social Behaviors, and Loneliness) and emotional well-being (Positive and Negative Affect and Depressive Symptoms) were tested. As shown in Table 4, NT demonstrated marginal to small positive relationships with socially anxious behaviors and Negative Affect. NE showed a medium positive relationship with empathetic concern and a marginal positive relationship with Positive Affect. PT demonstrated small relationships with Empathetic Concern and Positive Affect. PE demonstrated small to medium positive correlations with empathetic concern, Perspective-Taking, and Positive Affect, and a small negative relationship with Loneliness.

## 4. Discussion

This study aimed to investigate the optimal factor structure of the IRQ using a Chinese youth sample and to examine how IER, tested by C-IRQ, could predict young people’s social and emotional well-being. Based on the IER theory [6] and the measurement model of IRQ [9], we examined the factor structure of the C-IRQ, employing an ESEM approach. The findings showed that the four-factor structure best represented the current data, supporting the conceptualization of IER as a multidimensional construct in a non-western culture. Specifically, the four-factor structure of IER containing NT, PT, NE, and PE was supported by our factor analyses. Although the correlations between IRQ scores and social and emotional well-being indicators were weaker than expected, this study provides preliminary evidence of the convergent and discriminant validity of the IRQ in a non-Western culture. The validation and use of the IRQ can provide some insight into the growing research of IER.

Five alternative models were estimated together with the original four-factor structure model (M0). Compared with the four-factor model, the alternative models (except the one general factor model M1) showed comparable model fits. However, the factor loadings for the second-order factor models (M2, M4, and M5) and composite reliability for the bi-factor model (M3) are not sufficient for retaining these alternative models. The four-factor structure model demonstrated the best psychometric properties (adequate model fit indices, item loadings, and internal consistencies), consistent with the factor structure originally developed by Williams using the American sample [9]. Nevertheless, the item loadings on target factors and the inter-factor correlations were slightly lower than the values reported in previous studies [9]. One reason might be that using emotional suppression was significantly positively correlated with interpersonal harmony among Chinese but not European Americans [53]. When either positive or negative emotional events occur, sharing emotions is not a prioritized strategy for people in a society that values interpersonal harmony.

We established strict measurement invariance across gender groups for the optimal four-factor model, supporting the latent mean level comparison between female and male groups [54]. In this sample, the NT reported by women was significantly higher than by men, which is consistent with previous findings presented by Ding et al. and Liu et al. [20,55]. Inconsistent with Ding et al.’s study [20], women reported higher NE scores than men. The reason for this inconsistency might be the age differences of samples (M_age_ = 14.52 vs. M_age_ = 19.59 years old) used in these two studies. With age and experience, women’s efficacy in sharing negative emotions increases significantly. Another possible explanation is that men suppress their emotional expression more frequently compared with women [56,57].

The findings showed that PT and PE were marginally associated with extraversion, supporting the notion that IER is a different structure that captures richer content underlying social behaviors, rather than simply enjoying social interaction [9]. Inconsistent with the previous study suggesting a positive correlation between IER scores (except NT) and effective use of emotion regulation strategies (reappraisal) [9], people reporting high scores in using IER (except PE) are associated with low intrapersonal ER resources. However, these findings were somewhat consistent with previous studies that suggested positive associations between emotion regulation difficulties and IER strategies [17,34]. It may be the case that individuals with low intrapersonal emotion regulation ability (low inner emotion regulation resources) prefer to seek external resources to regulate their emotions.

Consistent with Williams et al.’s study [9], IRQ subscales (except NT) present positive associations with Empathetic Concern. People with high Empathetic Concern traits may anticipate the feelings of receivers and therefore make a more considered decision before seeking external resources. We detected a positive relationship between NT and Anxious Social Behaviors, which was consistent with previous studies presenting positive relationships between IER strategies and general anxiety symptoms [17,27,32]. For emotional well-being, the IRQ subscales (except NT) present a positive correlation with Positive Affect, although these correlations were weaker than formerly reported [9,20,34]. A significant positive relationship between NT and Negative Affect was detected in the current sample, but no such significant relationship was found in previous American samples [9] or Chinese adolescent samples using the IRQ [20]. One reason might be that individuals experiencing greater negative affect are more likely to use external resources to regulate their emotions [17]. Another explanation might be that the emotional support receipt increases individuals’ emotional costs, which in turn leads to more negative emotional symptoms [10]. Overall, the correlations between IRQ scores and social and emotional well-being indicators were weaker than expected. This may be because interpersonal behaviors have accelerated the spread of pandemic anxiety and bad news, which has diminished the positive effects of interpersonal emotional regulation behaviors.

## 5. Limitations and Conclusions

The current study has several limitations worth noting. First, the sample was overrepresented by female students; a low proportion of males may limit the explanation of the latent gender difference on IER. Future studies could note the gender difference in responding to questionnaire surveys and encourage the participation of males. Second, some brief measures showed relatively low reliability (Extraversion, Perspective-Taking, and Empathetic Concern). Future research should replicate the current scales with more representative and reliable measures. Third, the participants responded to the questionnaires during the intermittent restrictions of COVID-19, which may have limited participants’ reports of their IER behaviors, especially for people favoring face-to-face interactions. With the normalization of the prevention and control of COVID-19, further replication studies could consider the effects of the COVID-19 restrictions on movement and interaction on people’s IER behaviors. Finally, we did not include clinical samples in the validating process. Future research could examine the structure of IER using both population and clinical samples to further explore the possible applications of IER in clinical interventions for affective disorders. In conclusion, the Chinese version of the IRQ (C-IRQ) have presented adequate psychometric properties and would be a useful tool for measuring interpersonal emotion regulation behaviors.

## Figures and Tables

**Figure 1 behavsci-13-00507-f001:**
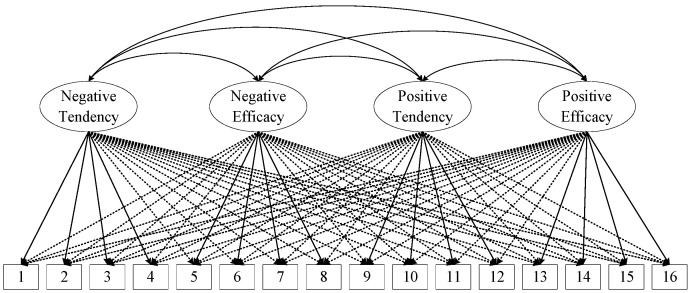
Graphical representation of the four-factor exploratory structural equation model.

**Figure 2 behavsci-13-00507-f002:**
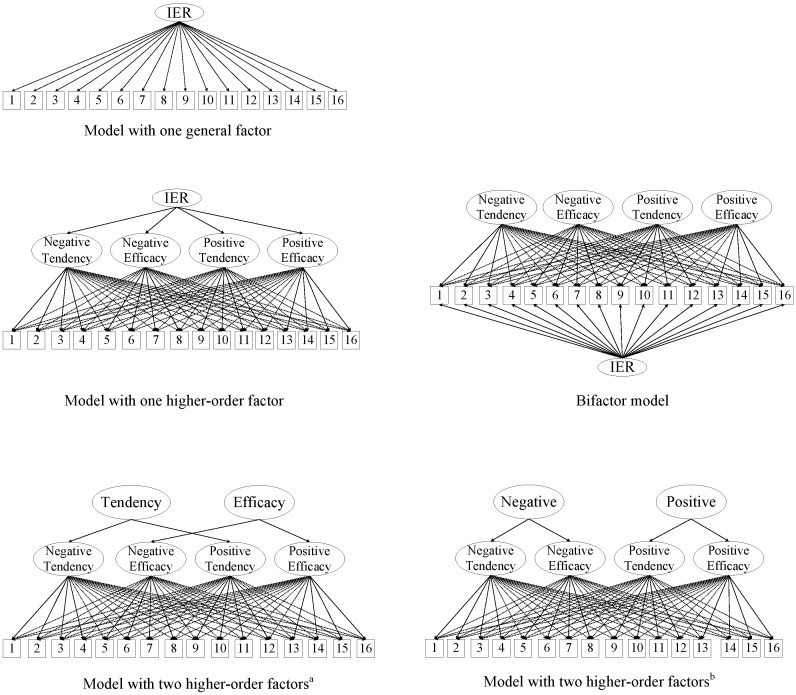
Graphical representation of the alterative exploratory structural equation models examined in the study. *Note.* IER = interpersonal emotion regulation; ^a^ two higher-order factors are Tendency and Efficacy; ^b^ two higher-order factors are Negative and Positive.

**Table 1 behavsci-13-00507-t001:** Factor loadings and inter-factor correlations for the four-factor ESEM model.

	Items	NT	NE	PT	PE
1.	When something bad happens, my first impulse is to seek out the company of others.	**0.843**	0.109	−0.085	−0.022
2.	When I’m having trouble, I can’t wait to tell someone about it.	**0.831**	0.184	−0.114	−0.006
3.	I just have to get help from someone when things are going wrong.	**0.403**	−0.242	0.234	0.154
4.	I manage my emotions by expressing them to others.	**0.504**	−0.187	0.187	−0.011
5.	I appreciate having others’ support through difficult times.	−0.045	**0.702**	0.002	0.037
6.	Sometimes I just need someone to understand where I’m coming from.	0.069	**0.691**	0.094	0.003
7.	It really helps me feel better during stressful situations when someone knows and cares about what I’m going through.	0.094	**0.629**	0.071	0.055
8.	I really appreciate having other people to help me figure out my problems.	0.013	**0.592**	0.162	0.085
9.	When things are going well, I just have to tell other people about it.	0.057	0.098	**0.744**	−0.022
10.	When something good happens, my first impulse is to tell someone about it.	−0.130	0.126	**0.797**	0.021
11.	When things are going well, I feel compelled to seek out other people.	0.119	−0.156	**0.769**	−0.040
12.	When I want to celebrate something good, I seek out certain people to tell them about it.	0.074	0.268	**0.436**	0.107
13.	I’m happier when I’m with my friends than when I’m by myself.	−0.048	−0.003	−0.030	**0.740**
14.	Being with other people tends to put a smile on my face.	−0.058	0.100	−0.062	**0.794**
15.	I find that even just being around other people can help me to feel better.	0.155	−0.169	0.038	**0.663**
16.	I really enjoy being around the people I know.	−0.005	0.088	0.023	**0.644**
	Latent inter-factor correlations ^a^	NT	NE	PT	PE
	Negative Tendency (NT)	1			
	Negative Efficacy (NE)	0.14	1		
	Positive Tendency (PT)	0.57	0.24	1	
	Positive Efficacy (PE)	0.43	0.36	0.46	1

*Note.* ESEM = structural equation modeling; ^a^ all correlation coefficients are significant at 0.01 level.

**Table 2 behavsci-13-00507-t002:** Summary of model fit indices in competing ESEM models.

Model	Description	S-B Scaled χ^2^	*df*	CFI	RMSEA [90% CI]	SRMR	ΔCFI	ΔRMSEA
M0	Model with four factors	183.10	62	0.950	0.060 [0.050–0.070]	0.027		
M1	Model with one general factor	1140.58	104	0.573	0.136 [0.129–0.143]	0.115		
M2	Model with one higher-order factor	183.95	63	0.950	0.060 [0.050–0.070]	0.027		
M3	Bifactor model	127.02	50	0.968	0.053 [0.042–0.065]	0.020		
M4	Model with two higher-order factors ^a^	181.00	62	0.951	0.060 [0.050–0.070]	0.027		
M5	Model with two higher-order factors ^b^	183.15	62	0.950	0.060 [0.050–0.070]	0.027		
Invariance across gender								
	Configural invariance	262.59	124	0.944	0.064 [0.054–0.075]	0.031		
	Metric invariance	325.42	172	0.938	0.058 [0.048–0.067]	0.047	−0.006	−0.006
	Scalar invariance	348.85	184	0.934	0.058 [0.048–0.067]	0.049	−0.004	<0.001

*Note.* ESEM = structural equation modeling; S-B scaled χ^2^ = Satorra–Bentler chi-square; *df* = degrees of freedom; CFI = comparative fit index; RMSEA = root mean square error of approximation; 90% CI = 90% confidence interval for the RMSEA; SRMR = standardized root mean square residual; ^a^ two higher-order factors are Tendency and Efficacy; ^b^ two higher-order factors are Negative and Positive.

**Table 3 behavsci-13-00507-t003:** Standardized factor loadings for alternative models.

Item		Model with One General Factor				Bifactor Model							Model with Two Higher-Order Factors (Tendency and Efficacy)						Model with Two Higher-Order Factors (Positive and Negative)			
	HF	NT	NE	PT	PE	FG	NT	NE	PT	PE	HF1	HF2	NT	NE	PT	PE	HF1	HF2	NT	NE	PT	PE
NT	**0.27**										**0.05**						**0.18**					
Item 1		**0.698**	0.108	0.304	−00.021	0.605	**0.415**	0.009	−0.013	−0.020			**0.705**	0.109	0.267	−0.021			**0.692**	0.390	0.108	−0.021
Item 2		**0.682**	0.169	0.274	0.001	0.579	**0.770**	0.075	0.003	0.021			**0.695**	0.183	0.235	−0.006			**0.676**	0.358	0.169	0.001
Item 3		**0.348**	−0.241	0.410	0.152	0.717	**−0.055**	−0.264	−0.087	−0.062			**0.337**	−0.241	0.400	0.152			**0.345**	0.453	−0.241	0.151
Item 4		**0.427**	−0.190	0.409	−0.009	0.569	**0.156**	−0.224	0.007	−0.121			**0.422**	−0.186	0.392	−0.011			**0.423**	0.460	−0.190	−0.009
NE	**0.37**											**0.37**					**0.21**					
Item 5		−0.037	**0.699**	0.002	0.036	0.115	0.082	**0.652**	0.131	0.161			−0.038	**0.698**	0.002	0.036			−0.037	0.002	**0.699**	0.036
Item 6		0.032	**0.686**	0.160	0.004	0.356	−0.002	**0.680**	0.041	0.032			0.058	**0.687**	0.140	0.003			0.032	0.168	**0.685**	0.004
Item 7		0.055	**0.625**	0.147	0.054	0.419	−0.051	**0.649**	−0.046	0.026			0.079	**0.626**	0.127	0.054			0.054	0.158	**0.624**	0.054
Item 8		−0.008	**0.590**	0.199	0.084	0.352	0.039	**0.558**	0.130	0.122			0.011	**0.589**	0.185	0.084			−0.008	0.202	**0.590**	0.084
PT	**0.47**										**0.10**							**0.52**				
Item 9		0.049	0.110	**0.774**	−0.027	0.606	0.010	0.090	**0.506**	−0.019			0.048	0.098	**0.769**	−0.021			0.048	**0.781**	0.110	−0.027
Item 10		−0.108	0.125	**0.746**	0.021	0.498	−0.048	0.134	**0.604**	0.050			−0.109	0.125	**0.747**	0.021			−0.107	**0.734**	0.125	0.021
Item 11		0.110	−0.142	**0.813**	−0.045	0.621	−0.011	−0.149	**0.464**	−0.085			0.099	−0.155	**0.813**	−0.039			0.109	**0.825**	−0.142	−0.044
Item 12		0.057	0.273	**0.486**	0.103	0.470	0.082	0.256	**0.385**	0.146			0.062	0.267	**0.477**	0.106			0.057	**0.496**	0.273	0.103
PE	**0.93**											**0.93**						**0.83**				
Item 13		−0.029	−0.002	−0.033	**0.728**	0.345	0.025	0.081	0.059	**0.624**			−0.040	−0.003	−0.027	**0.729**			−0.029	−0.030	−0.002	**0.725**
Item 14		−0.048	0.099	−0.058	**0.782**	0.386	0.012	0.191	0.026	**0.662**			−0.049	0.099	−0.058	**0.783**			−0.048	−0.057	0.099	**0.779**
Item 15		0.147	−0.168	0.118	**0.653**	0.567	−0.005	−0.091	−0.048	**0.450**			0.130	−0.168	0.120	**0.654**			0.145	0.140	−0.168	**0.650**
Item 16		0.003	0.088	0.043	**0.633**	0.452	−0.058	0.153	0.011	**0.497**			−0.004	0.087	0.044	**0.635**			0.003	0.049	0.088	**0.631**

*Note.* NT = Negative Tendency, NE = Negative Efficacy, PT = Positive Tendency, PE = Positive Efficacy; HF = Higher-order Factor; FG = General Factor.

**Table 4 behavsci-13-00507-t004:** Correlations between the Interpersonal Regulation Questionnaire (IRQ) subscales and other variables.

			Social Well-Being				Emotional Well-Being		
IRQ Scales	Extra-Version	Emotion Dysregulation	Empathetic Concern	Perspective Taking	Social Anxious Behaviors	Loneliness	Positive Affect	Negative Affect	Depressive Symptoms
Negative Tendency	0.05	0.18 **	0.09	−0.05	0.10 †	0.05	0.10	0.15 *	0.04
Negative Efficacy	0.08	0.14 *	0.26 **	0.09	0.06	−0.02	0.11 †	−0.05	−0.05
Positive Tendency	0.11 †	0.19 **	0.14 *	0.02	0.05	0.00	0.13 *	0.04	0.04
Positive Efficacy	0.11 †	−0.07	0.20 **	0.25 **	−0.05	−0.17 **	0.25 **	0.06	−0.08
Total (IRQ)	0.13 *	0.13 *	0.24 **	0.11 *	0.05	−0.06	0.21 **	0.07	−0.02

*Note.* † *p* < 0.08; * *p* < 0.05; ** *p* < 0.01.

## Data Availability

Restrictions apply to the availability of these data. Data were obtained from the Henan Provincial Philosophy and Social Science Planning Office and are available from the contact authors with the permission of the Henan Provincial Philosophy and Social Science Planning Office.

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
