# Peer review of "Psychometric Properties of Interpersonal Regulation Questionnaire for Chinese College Students: Gender Differences and Implications for Well-Being"

_behavsci, 2023, doi:10.3390/bs13060507_

Round 1

Reviewer 1 Report

The authors were examing the factor structure of a newly translated Chinese IRQ scale using a College sample. Backwards translation was employed in creating the scale.

The quality of Enlish in the article was strong.

Unfortunately, I could not open the supplementary file. From what the authors say in Lines 246-7 I gather that it contains additional information on model fit.

The statistical treatment in the article was relatively complex: I am not an authority on ESEM but I was able to follow the authors' descriptions most of the time. I note that the authors were comparing several different approaches to envisioning the factor structure of the Chinese IRQ.

Here are my remarks:

1. The reference to "all measures" in line 159 is unclear as these measures have not been introduced.

2. Line 174. Extroversion is measured using a two item scale with a relatively low inter-item correlation (.54). How useful is this scale, really? Its low correlations with IRQ subscales might be due to its weakness.

3. Lines 161-2. The authors refer to what seems to be informed consent, but do not mention an ethical review or ethical approval from their institution.

4. Line 189. The Cronbach's alphas for the IRI-C are quite low.

5. Table 4. The equivalence between variables in this table and those listed in the Method section "Measures" is unclear. Were only some variables used? If so, why mention the rest? 

6. Correlations in Table 4 are suprisingly weak. The authors do discuss this, but they have not resolved the problem. With n>500, one should not have to resort to discussing outcomes where p<.08. As well, interpretations of the relationships in Table 4 occupy a good deal of the discussion section. 

7. The authors provide a mass of data for evaluating various models. They provide helpful drawings for the models, but the comparative discussion of which model has a better fit is not clearly summarized in one place. In the Discussion (line 344) the four-factor structure was deemed the best, but this conclusion is not supported by any specific arguments. Table 2 suggests that several different models had comparable fit indexes. 

Conclusion: If the above issues can be addressed, this article would be easier for most readers to understand.

Reviewer 2 Report

This is an article that aims to analyze the psychometric characteristics of the questionnaire in order to establish a model that makes it possible to predict well-being in the Chinese student population. The article is rigorous and good at a methodological level, but on the one hand it requires including unspecified information, and on the other it presents serious formatting problems that must be corrected.

Abstract.

The section far exceeds the maximum limit indicated by the magazine, 270 words against 200. It is also sectioned with titles, when otherwise specified in this magazine. In addition, it would be recommendable that the authors clarify and specify it more, since at the beginning it wanders too much, never exceeding 200 words.

Introduction:

In my opinion, the introduction provides the appropriate theoretical basis for the work at hand, but formally it has serious problems with the number of references. The reference numbers must be adjusted to the format of the journal. Citation numbers must be between square brackets and before the punctuation. In the text they appear without square brackets, in superscript, and after the punctuation. Example: Line 35, citations 1,2 are without brackets, in superscript and after the point. The whole document is like this. Review and modify entire.

Method:

Sample:

In this section, very important information is missing. The authors present data such as the mean age of the sample and the standard deviation, but in this type of study more relevant information is required, such as:

How was the selection of the sample? Random, incidental, etc... It only specifies that they were "recruited".

Was the ideal sample size for this study statistically calculated in some way?

There is talk of students between 17 and 31 years old, with the average age being 19.59 years. It would be very good to reflect the number of participants by age in a validation process like this.

Measures: Several instruments do not review their authors, or authors who validated these instruments for the Chinese population. This must be corrected.

results:

The figures should be more centered according to the format of the document.

Discussion:

The discussion is generally correct, but it still has serious formatting errors in terms of the numbering of references required by the journal. Exactly the same ones that have been reflected in the introduction section. Its modification is necessary.

References:

The format of the references is wrong. To begin the references section has incorrect line spacing. It must be corrected.

Some example of reference with incorrect format, although it is a general problem of the whole section.

Eg: Reference 1 (In red what is missing).

  1. Coan, J.A.; Schaefer, H.S.; Davidson R.J. Lending a hand: social regulation of the neural response to threat. Psychol Sci. 2006;17(12):1032-1039. doi:10.1111/j.1467-9280.2006.01832.x

References:

The format of the references is wrong. To begin the references section has incorrect line spacing. It must be corrected.

Some example of reference with incorrect format, although it is a general problem of the whole section. Eg: Reference 1 of scientific article (In red what is missing).

As a general guideline, punctuation marks are missing from all references. There are no commas between the last name and the first name. Dots are missing between the initials. There is a semicolon between authors. The year of publication should be in bold. The volume of the magazine should be in italics. The name of the journal should not be abbreviated. These errors keep happening. 

In other references there are even more errors, for example in reference number 24, the journal is not in italics either and this happens in many others (References 30, 33, etc.). In book references (eg: 46), do not italicize the title of the book.

Authors are recommended to review the section and do it correctly. It is recommended to review the journal's standards regarding references.

Round 2

Reviewer 2 Report

All changes was made. For me the paper it's ok for publication.

I would only modify lines 152 and 153. I would leave only the reference number [56], or at least remove the year of publication.

Good job. Congratulations.